Inhibition of Kelch-like epichlorohydrin-related protein 1 promotes the progression and drug resistance of lung adenocarcinoma

Gao Hong 1
Tang Peipei 2
Ni Kejie 1
Zhu Lun 3
Chen Song 2
http://orcid.org/0000-0002-1401-9883 Zheng Yulong 1 ha183@163.com
Wan Yufeng 1 ggwanyufeng@163.com
1 Department of Respiratory Diseases, The Affiliated Huai’an Hospital of Xuzhou Medical University , Huai’an, Jiangsu , China
2 Institute of Medicinal Biotechnology, Jiangsu College of Nursing , Huai’an, Jiangsu , China
3 Department of Pathology, The Affiliated Huai’an Hospital of Xuzhou Medical University , Huai’an, Jiangsu , China
Uversky Vladimir
Electronic publication date: 2021 Aug 19
Publication date: 2021
Volume: 9
Electronic Location ID: e11908
Received 2021 Jan 12; Accepted 2021 Jul 14
Copyright: © 2021 Gao et al.
Copyright year: 2021
Copyright holder: Gao et al.
License: This is an open access article distributed under the terms of the Creative Commons Attribution License, which permits unrestricted use, distribution, reproduction and adaptation in any medium and for any purpose provided that it is properly attributed. For attribution, the original author(s), title, publication source (PeerJ) and either DOI or URL of the article must be cited.
License URL: https://creativecommons.org/licenses/by/4.0/

Keywords: Keap1, LUAD, Progression, Drug resistance

Funding: Natural Science Foundation of Huaian, Jiangsu, China HAB202073 and HABZ201808 The current study was supported by grants from the Natural Science Foundation of Huaian, Jiangsu, China (No. HAB202073 and No. HABZ201808). The funders had no role in study design, data collection and analysis, decision to publish, or preparation of the manuscript.

==============================
Background

Lung cancer is a common malignant carcinoma of respiratory system with high morbidity and mortality. Kelch-like epichlorohydrin-related protein 1 (Keap1), a member of the BTB-Kelch protein family, has been reported as an important molecule in several cancers. However, its potential role in tumor is still controversial. Here we aim to clarify the effect of Keap1 on the biological characteristics and chemotherapy resistance in lung adenocarcinoma (LUAD).

Methods

Immunohistochemistry was conducted to compare Keap1 expression in lung adenocarcinoma tissues and matched non-cancerous tissues, and the correlation between Keap1 expression and clinicopathological features was analyzed. Subsequently, the stable A549 and H1299 cell lines with Keap1 knockdown or overexpression were constructed using lentivirus. The roles of Keap1 on the cell proliferation, migration, invasion and drug resistance were investigated by colony formation assay, cell proliferation assay, wound scratch test, transwell invasion assay and drug sensitivity assay, respectively.

Results

Keap1 was lowly expressed in tumor tissues compared to matched non-cancerous tissues, and its expression was correlated with TNM stage and lymph node metastasis. Early stage (I) tumors without lymph node metastasis had higher levels of Keap1 expression compared with late-stage tumors (II, III) with the presence of lymphatic metastasis. Colony formation assays showed that Keap1 knockdown promoted the proliferation of A549 and H1299 cells, and the cell growth curves further confirmed this feature. In contrast, wound scratch and transwell invasion experiments showed that Keap1 overexpression inhibited cell migration and invasive malignancy. The IC50 for cisplatin and paclitaxel were significantly increased by Keap1 knockdown in A549 and H1299 cell lines.

Conclusion

Keap1 knockdown promotes tumor cell growth, proliferation, invasion, metastasis and chemotherapy resistance in LUAD. It may be a potential tumor marker to guide the staging and treatment of lung cancer.

Introduction

As the world’s largest cancer killer, lung cancer accounts for approximately 2.09 million new cases (11.6% of all new cancer cases) and 1.77 million lung cancer-related deaths (18.4% of all cancer deaths) each year. Its mortality and incidence rates rank first among all cancers (Bray et al., 2018). LUAD is the most common form of non-small-cell lung cancer, and mostly at advanced stage when diagnosed (Cao & Chen, 2019). Kelch-like epichlorohydrin-related protein (Keap1) protein belongs to the BTB-Kelch protein family. The BTB-Kelch protein family mainly assembles with Cullin3 and Rbx1 to form multi-subunit Cullin-RING ligases for protein ubiquitination (Dhanoa et al., 2013). Keap1 comprises five domains: N-terminal region (NTR), Broad complex, Tramtrack and Bric-à-Brac (BTB), intervening region (IVR), double glycine repeat (DGR) and C-terminal region (CTR). The BTB region acts as a substrate-specific adaptor for Cullin3 ubiquitin ligase, which mediates the homodimerization of Keap1 and the subsequent constitutive ubiquitination of Nrf2. The IVR region contains cysteine residues which are sensitive to oxidation, or to covalent modification by electrophiles. The DGR and CTR regions are also called DC domains, mainly mediating the interaction with and Nrf2 (Ogura et al., 2010; Canning, Sorrell & Bullock, 2015).

The most important function of Keap1 in the body is to play an anti-oxidative stress effect by the Keap1-nuclear factor E2-related factor2 (Nrf2) pathway. Under steady-state conditions, Keap1 binds to Nrf2 and anchors it in the cytoplasm. At the same time, it assembles with the Cullin3 protein to form a Cullin-RING E3 ligase complex for the degradation of Nrf2 by the 26S proteasome, so that Nrf2 is in a non-free and continuously degraded non-free state (Canning, Sorrell & Bullock, 2015). Keap1 protein contains a total of 27 cysteine residues (Cys), which are vulnerable to reactive oxygen species (ROS), or to covalent modification by electrophiles, especially the Cys151, Cys257, Cys273, Cys288, Cys297, Cys434 and Cys6 (Luo et al., 2007). This characteristic determines the high redox sensitivity of Keap1. Under conditions of oxidative stress, or the presence of electrophilic xenobiotics, Keap1 cysteine residues are oxidized, thereby altering the Cullin-RING E3 ligase complex and losing the Keap1-Nrf2 interaction, ultimately leading to Nrf2 translocation and nuclear accumulation. In the nucleus, the actived Nrf2 forms a heterodimer with the small Maf proteins, which belongs to Maf family of transcription factors (Tsuchiya et al., 2015). The heterodimer recognizes antioxidant response element that regulates the downstream transcription of target genes, which encode proteins acting as redox balancing factors, detoxifying enzymes, stress response proteins and metabolic enzyme, thereby exerting cytoprotective effects (Bellezza et al., 2018).

However, as one of the main factors regulating oxidative stress in the body, various recent studies have shown that Keap1 mutations are associated with cancers. Schulze et al. (2015) found Keap1 mutations in about 14% of hepatocellular carcinomas by exon sequencing. Chu et al. (2018) reported Keap1 heterozygous missense mutations in cervical cancer. Besides, Keap1 mutations have also been reported in lung cancer. An earlier study reported a mutation rate of Keap1 in lung squamous cell carcinoma is about 12% (Meyerson, 2012). While recently, a large-scale genomic exon sequencing study including 660 cases of lung adenocarcinoma (LUAD) and 484 cases of lung squamous carcinoma found that Keap1 gene was significantly mutated only in LUAD (Frank et al., 2018). However, Keap1 expression in solid tumors is variable and correlate with clinicopathological features. Researchers found that Keap1 protein was highly expressed in both endometrial and pancreatic cancers by immunohistochemistry (IHC), and that high levels of Keap1 protein expression were associated with poor prognosis in endometrial cancer (Ahtikoski et al., 2019; Isohookana et al., 2015). While another investigators reported that Keap1 protein was lowly expressed in breast cancer and osteosarcoma tissues and correlated with clinicopathological features such as lymph node metastasis (Zhang et al., 2016b; Zhang et al., 2016a). These studies suggest that the abnormal status of Keap1 in the body may influence tumorigenesis and progression.

So, in this study, we aimed to identify the effect of Keap1 on the biological characteristics and chemoresistance in LUAD, and to provide a new entry point for clinical diagnosis and targeted therapy.

Materials & Methods

Patient samples

The Institutional Review Board of the Affiliated Huai’an Hospital of Xuzhou Medical University (HEYLL201920) approved this study and all subjects signed an informed consent form prior to the study. Lung cancer specimens for this study were collected retrospectively from 37 patients in the hospital from January 2016 to December 2018. Matched non-cancerous tissue was defined as >5–10 cm from the cancerous tissue. The inclusion criteria were as follows: (1) Pathologically confirmed adenocarcinoma of the lung according to the “ICD-O-3 Hist/Behavior”; (2) All patients underwent radical resection without induction of chemotherapy or radiation therapy prior to surgery. (3) Patients did not have other malignancies prior to diagnosis; (4) Re-staging TNM staging according to the American Joint Committee on Cancer 8th edition TNM staging based on original staging and tumor size provided (Detterbeck et al., 2017). The exclusion criteria were as follows: (1) Patients with comorbid acute cardiovascular and autoimmune diseases; (2) Patients with missing or unknown clinical information.

Immunohistochemistry assay

The IHC was carried out according to standard protocol. In shortly, all paraffin specimens were sliced to a thickness of 4 µm, then deparaffinized and hydrated and heated in Tris/EDTA buffer pH 9.0 for antigen retrieval. The Keap1 antibody (1:500, ab227828, Abcam) was incubated overnight at 4 °C and the specimens were visualized with a light microscope (Olympus, Tokyo, Japan). Two pathologists completed the readings independently. The average number of positive cells in each field was analyzed as the percentage of positive cells in the section and scored using the Fromowitz criteria (Fromowitz et al., 1987): 0–5% was scored as 0, 6–25% was scored as 1, 25–50% was scored as 2, 51–75% was scored as 3, and >75% was scored as 4. The intensity of staining was scored according to the staining characteristics of most positive cells: 0 score for no staining, 1 score for light yellow, 2 score for brown, and 3 score for dark brown. The final score was derived by summing the percentage of stained cells and the corresponding intensity, with 0–1 being classified as negative (−), 2–3 as weakly positive (+), 4–5 as moderately positive (++), and 6–7 as strongly positive (+++). Where (−) and (+) recorded as low expression, (++) and (+++) recorded as high expression groups.

Cell culture and transfection

The human bronchial epithelial HBE cell line, human lung cancer cell lines NCI-A549, NCI-H1299, NCI-H460, NCI-H2126 and human embryonic kidney HEK-293FT cell lines were obtained from Huai’an Key Laboratory of Gastrointestinal Cancer. The HBE, NCI-H460 and NCI-H2126 cell lines were cultured in RPMI 1640 medium (Gibco, Waltham, MA, USA) with 10% fetal bovine serum (FBS; Gibco, Waltham, MA, USA) and incubated in a moist air incubator with 5% CO2 at 37 °C. The NCI-A549, NCI-H1299 and HEK-293FT cell lines were maintained in Dulbecco’s Modified Eagle’s Medium (DMEM; Gibco, Waltham, MA, USA) containing 10% FBS in 5% CO2 at 37 °C. Fresh medium was replaced every 1–2 days. The cells were passaged in a dilution ratio of 1:4 approximately every two days when the confluency reached about 80%. The cDNA sequence of Keap1 gene (GenBank accession number NM_203500) was cloned into Lenti-EFs-Flag-puro vector for Keap1 overexpression. Short hairpin RNA (shRNA) sequences targeting Keap1 was constructed into pLKO.1 lentiviral vector for Keap1 knockdown. For lentiviral packaging system, we incubated HEK-293FT cells in six-well plate and when the cell density reached 90% or higher, the VSVG envelope plasmid, Gag/Pol packaging plasmid, Keap1 overexpression plasmid, Keap1 knockdown plasmid and the control plasmid were transfected into HEK-293FT cells according to the Lipo8000™ transfection kit (Beyotime Biotechnology, Shanghai, China). The virus particles were collected after 48 h. For construction of stable cell lines, the A549 and H1299 cell lines were seeded in 24-well plate and when fusion reached nearly 30%, appropriate amounts of virus were added. After 24 h, we replaced the fresh medium, waited for the cell density to reach approximately 80%, and then added 2 μg/ml puromycin (Sigma, St. Louis, MO, USA) to screen the stable cell lines.

Western blot analysis

Protein lysates containing DL-Dithiothreitol (DDT) and Sodium dodecyl sulfate (SDS) were used to extract total proteins. The cytosolic and nuclear fractions of Nrf2 were prepared using a Nuclear and Cytoplasmic Protein Extraction Kit (Beyotime Biotechnology, China). Protein quantification was performed using BCA protein quantification kit (Beyotime Biotechnology, China). The volume that contains 20 μg protein was calculated to be taken as samples for 8%–13% sodium dodecyl sulfate polyacrylamide gel electrophoresis (SDS-PAGE). Then the gel was transferred to a nitrocellulose (NC) membrane with constant voltage at 30 V in an ice-water bath for overnight. After transferring the membrane, it was blocked with 5% nonfat-dried milk for 1 h at room temperature. The NC membrane was then incubated with primary antibody for 4 h at 37 °C and the appropriate HRP-conjugated secondary antibody was further incubated for 1h at room temperature. The Enhanced chemiluminescence (Beyotime Biotechnology, China) reagents A and B were mixed thoroughly in a 1:1 ratio, incubated with NC membrane for 1 min, and then exposed the NC membrane to the dark chamber of a chemiluminescence imaging system (Azure US Biosystems). β-actin was as an internal control. Image J software was applied for the quantification of protein. The primary antibodies used in western blotting were listed in Table 1.

Table 1 The primary antibody used in western blot.

Name	Product number	Company	
Keap1	ab227828	Abcam	
Nrf2	ab62352	Abcam	
E-cadherin	#3195	Cell Signaling	
N-cadherin	ab76011	Abcam	
Vimentin	10366-1-AP	Proteintech	
Snail	ab216347	Abcam	
β-actin	20536-1-AP	Proteintech	
LaminB	ab16048	Abcam	

Reverse transcription-quantitative polymerase chain reaction (RT-qPCR)

Total RNA was extracted using Trizol reagent (Invitrogen, USA). The integrity of RNA was tested by agarose gel electrophoresis and competent RNA produced clear 28S and 18S rRNA bands after processing. The quality of RNA was tested using a spectrophotometer/fluorometer (DeNovix, USA), only RNA samples with a 260/280 ratio between 1.8 and 2.2 were used in reverse transcription to synthesize cDNA using the PrimeScript™ RT reagent Kit with gDNA Eraser reverse transcription kit (Takara, Japan). Then, the fluorescent quantitative PCR reaction system was analyzed for 100 ng of cDNA template following the protocol of TBGreen®Premix Ex Taq™ kit (Takara, Japan). The specific PCR amplification procedure was as follows: 95 °C for 30 s, then 95 °C for 5 s, 60 °C for 30 s with 40 cycle. The 3-glyceraldehyde-phosphate dehydrogenase (GAPDH) was applied as an endogenous reference. Relative quantitative analysis using 2–∆∆Ct method. The primer sequences were as following: GAPDH-F: 5’-ACT TTG GTA TCG TGG AAG GAC TC-3’, GAPDH-R: 5’GAG GCA GGG ATG ATG TTC TGG-3; Keap1-F: 5’-GCT GTG TGG AGT TGC ACC AGC GTG CC-3’, Keap1-R: 5’-CGT GGA AGA CCT CGG ACT CGC AGC GC-3’.

Colony formation assay

Cells in logarithmic growth phase were inoculated 500–1000 cells per well in six-well plates, and maintained at 37 °C in an atmosphere of humidified air with 5% CO2. The medium was replaced with fresh medium every 5 days. After ten days later, the six-well plates were taken out and the medium were removed. And the cells were fixed with 4% paraformaldehyde for 30 min, followed by staining with crystal violet for 10 min, then rinsed slowly and air-dried. Pictures were taken in the darkroom of Gel luminescence Imaging System (Azure Biosystems, USA), and the cell colony numbers were counted using Image-Pro Plus software.

Wound scratch test

Cells in logarithmic growth phase were spread in six-well plates at a density of 1 × 106 cells per well. When the cell confluence reached 95–100%, a 200ul yellow pipette tip was used to draw three vertical lines in each well and the exfoliated cells were washed away with PBS. Then, 2 ml of DMEM medium without FBS was added to each well and further incubated in a 37 °C, 5% CO2 incubator. Under the orthomosaic microscope (Olympus, Tokyo, Japan), the changes of the scratch width were observed and photographed (×40) at a certain time. The Image J software was used to quantitative analysis for the scratch area. Cell migration rate (%) = (initial scratch area – scratch area at specified time)/initial scratch area × 100%.

Transwell invasion assay

Transwell invasion assay was performed as previously described (Wang et al., 2017). In brief, an appropriate number of cells were seeded in the upper chamber of the 24-well Transwell chamber (Corning, New York, NY, USA), which containing 200 μl serum-free medium. A total of 600 μl of DMEM containing 20% FBS was plated in the bottom chamber. Following a 20 h incubation in 5% CO2 at 37 °C, the cells were rinsed, fixed with pre-cooled methanol for 30 min and stained with crystal violet for 15 min. The cells on the upper-side of the membrane were then removed with a clean cotton swab, and the cells on the under-side were observed and photographed under a light microscope (Olympus, Tokyo, Japan). The number of invaded cells was counted in three randomly selected fields using the Image-Pro plus software.

Cell proliferation assay

A total of 1 × 105 cells were seeded at 24-well plate. Each group of cells repeated in three parallel multiple wells. Then the cells were in culture medium at 37 °C in a 5% CO2 incubator, and digested with 0.25% trypsin (Gibco, Waltham, MA, USA) after 24, 48, 72 and 96 h, respectively. The digested cells were then counted with Count star automated cell counter (AlitLifeScience, Shanghai, China) and cell survival curves were plotted.

Drug sensitivity assay

The anti-tumor drug resistance on cells was measured by cell viability using the Cell Counting Kit-8 solution (CCK-8; MedChemExpress, Monmouth Junction, NJ, USA) as previously described (Zhang et al., 2018). Briefly, cells were seeded in 96-well plates at 5 × 103 cells/well in complete medium and incubated overnight. Then varying concentrations of cisplatin and paclitaxel (Qilu Pharmacy Co. Ltd., Jinan, China) were added into the medium and incubated for 24–48 h and 10 μl CCK-8 was added to each well for another 1 h. Triplicate wells were analyzed for each concentration. The absorbance was measured at 450 nm under a microplate reader (Bio-Rad, Hercules, CA, UAS). Cell survival rate (%) = [(As − Ab)/(Ac − Ab)] × 100, As = absorbance of experimental wells (absorbance of wells containing cells, culture medium, CCK-8 and treated with different concentrations of cisplatin), Ab = absorbance of blank wells (absorbance of wells containing medium and CCK-8), Ac = absorbance of control wells (absorbance of wells containing cells, medium and CCK-8). The half-maximal inhibitory concentration (IC50) was calculated from the rate of cell survival after normalization by the probit transformation.

Establishment of model of oxidative stress induced by H2O2

As previously described Sun et al. (2019), H2O2 (Sigma, St. Louis, MO, USA) was used to the establishment of a model of oxidative stress. In brief, the lung cancer cell lines, A549 cell lines with different levels of Keap1 were pre-treated with 200 µM H2O2 for 30 min and H1299 cell lines with different levels of Keap1 were pre-treated with 100 µM H2O2 for 30 min respectively.

Statistical analysis

SPSS software (version 26.0, USA) and GraphPad Prism software (version 8.0.2, CA) were used for statistical analysis. The measurement data were expressed as mean ± standard deviation (x ± SD), and unpaired t-test was used for comparison between groups. Count data were expressed as (%), and Fisher’s exact test was used to compare the clinical data and immunohistochemical expression. all statistical tests were two-sided probability tests. P < 0.05 values were considered significant.

Results

Keap1 was low expressed in lung adenocarcinoma (LUAD) tissue and cell lines

Thirty-seven cases of LUAD and matched non-cancerous tissues were detected by IHC assays. The result showed that Keap1 protein was mainly distributed in the cytoplasm, and rarely in nuclei (Figs. 1A and 1B). The extent of Keap1 expression was significantly higher in para-cancerous tissues than in LUAD tissue. Statistical analysis demonstrated that Keap1 was highly expressed in 67.6% of paraneoplastic tissues compared to 37.8% of LUAD tissues, and this difference was statistically significant (P = 0.019) (Table 2). Next, we further detected the expression levels of Keap1 protein in human lung cancer cell lines and human bronchial epithelial HBE cell line by western blot analysis and founded that lower expression of Keap1 in all cancer cell lines compared to HBE (Figs. 1C and 1D). More importantly, the fold change of Keap1 mRNA was significantly lower than that of HBE by RT-qPCR (Fig. 1E).

Figure 1 Keap1 was low expressed in LUAD tissue and cell lines.

(A–B) Representative immunohistochemical images with staining for Keap1 in different clinical stages of LUAD and matched non-cancerous samples. Photomicrographs were captured under ×400 magnification. Scale bar 20 μm. (A) Low Keap1 expression in LUAD (Stage III, left; Stage II, right); (B) High Keap1 expression in LUAD (Stage I, Left) and normal (Right). (C) Keap1 expression was tested and compared between human bronchial epithelial cell line HBE and lung cancer cell lines via Western blot analysis, β‐actin protein was served as an internal control. (D) Quantitative analysis of Keap1 protein in different cell lines was shown. (E) Expression of Keap1 in different lung cancer cell lines and human bronchial epithelial cells line HBE was compared by RT-qPCR. GAPDH was used as a reference for RNA. Statistical significance was tested by unpaired t test. Values were given as mean ± SD *P < 0.05, ** P < 0.01, ***P < 0.001 compared to the control group (HBE).

Table 2 Correlation of Keap1 with the occurrence of LUAD.

Group	n	Keap1	P	
Low expression (%)	High expression (%)		
LUAD	37	23 (62.2)	14 (37.8)	0.019	
Para-cancerous	37	12 (32.4)	25 (67.6)		

Keap1 low expression was mainly presented in advanced stage in LUAD tumors

We also analysed the association between Keap1 expression and clinicopathological features in the 37 LUAD patients, and noted that Keap1 high expression were more commonly observed in early stage (I) than in late-stage (II and III) patients (P = 0.030). Moreover, Keap1 expression was negatively correlated with lymph node metastasis, higher levels of Keap1 expression were more likely associate without lymph node metastasis (P = 0.017). But no correlation was observed between the expression levels of Keap1 and any other clinical parameters, including age, gender, smoking history, tumor location, differentiation and tumor max diameter (P > 0.05) (Table 3). Taken together, the above date indicated that down-regulation of Keap1 may be a critical event in tumor progression.

Table 3 Baseline characteristics of LUAD and the correlation with Keap1 expression.

Group	n	Keap1	P	
		Low expression (%)	High expression (%)		
Age (years)					
>63	16	12 (75.0)	4 (25.0)	0.315	
≤62	21	12 (57.1)	9 (42.8)		
Gender					
Male	19	14 (73.7)	5 (26.3)	0.313	
Female	18	10 (55.6)	8 (44.4)		
Smoking					
Yes	12	10 (83.3)	2 (16.7)	0.149	
No	25	14 (56.0)	11(44.0)		
Tumour site					
Left	15	11 (73.3)	4 (26.7)	0.491	
Right	22	13 (59.1)	9 (40.9)		
Differentiation					
Low	5	4 (80.0)	1 (20.0)	0.634	
Moderate	19	13 (68.4)	6 (31.6)		
High	13	7 (53.8)	6 (46.2)		
Stage					
I	15	6 (40.0)	9 (60.0)	0.030	
II	12	9 (75.0)	3 (25.0)		
III	10	9 (90.0)	1 (10.0)		
Lymph node metastasis					
Yes	16	14 (87.5)	2 (12.5)	0.017	
No	21	10 (47.6)	11 (52.4)		
Tumor max diameter					
>3.56 cm	16	12 (75.0)	4 (25.0)	0.315	
≤3.56 cm	21	12 (57.1)	9 (42.9)		

A549 and H1299 cell lines overexpressing and knocking down Keap1 were successfully constructed

Due to the low expression of Keap1 in LUAD patients and cell lines, we chose cells with the relatively high or low expression of Keap1 to more effectively investigate the phenotype changes. Then we stably knocked down and overexpressed Keap1 in A549 and H1299 cell lines using a lentiviral packaging system. To determine whether the A549 and H1299 cell lines overexpressing and knocking down Keap1 were successfully constructed, we examined the mRNA and protein expression levels of Keap1 by RT-qPCR and western blotting. As expected, compared with the control group, the Keap1 protein expression levels were significantly increased in the A549 and H1299 cell line overexpression group, while decreased in the knockdown group (Figs. 2A and 2B), RT-qPCR further validated the observations in western blotting assay (Fig. 2C), indicating that the Keap1 overexpression and knockdown A549 and H1299 cell lines were successfully constructed.

Figure 2 Keap1 protein and mRNA expression levels by western blot analysis and RT-qPCR.

(A) Expression of Keap1 protein level in A549 and H1299 cell lines was performed by western blotting analysis. β-actin protein was served as an internal control. (B) Quantitative analysis of Keap1 protein in A549 and H1299 cell lines was shown. (C) Expression of Keap1 mRNA level in A549 and H1299 cell lines was examined by RT-qPCR. GAPDH was used as a reference for RNA. Statistical significance was tested by unpaired t test. Values were given as mean ± SD. *P < 0.05, **P < 0.01, ***P < 0.001 compared to the control group.

Keap1 knockdown facilitated A549 and H1299 cell lines proliferation

To identified the impact of Keap1 on proliferation, we performed colony forming assay. Representative images revealed that cells with low Keap1 expression had an enhanced ability to form colonies, while cells with increased Keap1 expression levels formed fewer colonies compared to the control group (Fig. 3A). Quantification of the total number of colonies largely confirmed our observation (Fig. 3B). The result indicated that stable knockdown of Keap1 facilitated the proliferation of A549 and H1299 cells in vitro, and this effect could be further confirmed by cell growth curves up to 96 h (Fig. 3C).

Figure 3 Keap1 knockdown facilitated A549 and H1299 cell lines proliferation.

(A) Representative images of colony formation assay in A549 and H1299 cell lines expressing different levels of Keap1 were shown. (B) Quantitative analysis of total colony numbers in A549 and H1299 cells was shown. (C) Cell proliferation abilities in A549 and H1299 cell lines expressing different levels of Keap1 were analysed by cell growth curves. Statistical significance was tested by unpaired t test. Values were given as mean ± SD. **P < 0.01, ***P < 0.001 compared to the control group.

Keap1 knockdown promoted the migration and invasion in A549 and H1299 cell lines

The effect of Keap1 on the migration ability in A549 and H1299 cells was examined by scratch test. As shown in Figs. 4A and 4B, after drawing the lines at certain intervals, the width of the scratches gradually decreased in all cell groups, and the migration rate of the scratches were the highest in the Keap1 knockdown group, the second highest in the control group, and the lowest in the overexpression groups in A549 and H1299 cell lines. To investigate the role of Keap1 in LUAD cell invasion, transwell cell invasion assay was performed. The results showed that the number of cells passing through the membrane was significantly higher in the knockdown group A549 and H1299 cell lines than in the control group, whereas in the overexpressed group, the results were reversed (Figs. 4C and 4D).

Figure 4 Keap1 knockdown promoted the migration and invasion in A549 and H1299 cell lines.

(A) Wound scratch assay was performed with A549 and H1299 cell lines expressing different levels of Keap1. (B) Quantitative analysis of wound scratch assay in A549 and H1299 cell lines expressing different levels of Keap1. Images were captured at a certain time after wound was formed. The percentage of migration was assigned as 100% when complete fusion occurred, and 0% at t = 0 h. Relative migratory rates were shown in the graphs. (C) Transwell invasion assay of A549 and H1299 cell lines expressing different levels of Keap1. (D) Quantitative analysis of transwell invasion assay of A549 and H1299 cells was shown. Statistical significance was tested by unpaired t test. Values were given as mean ± SD. *P < 0.05, **P < 0.01, ***P < 0.001 compared to the control group.

Keap1 knockdown facilitated EMT of A549 and H1299 cell lines

To assess the role of Keap1 in tumor EMT, we performed western blot analysis of EMT marker proteins. The results showed that the expression of the epithelial marker E-cadherin was significantly elevated in A549 cells overexpressing Keap1, whereas the expression of mesenchymal markers (N-cadherin, Vimentin) and Snail was lower in cells overexpressing Keap1. In contrast, knockdown of Keap1 resulted in decreased expression of E-cadherin and increased expression of N-cadherin, Vimentin and Snail (Figs. 5A and 5B), which were consistent with that observed in H1299 cells (Figs. 5A and 5C).

Figure 5 Keap1 knockdown facilitated EMT of A549 and H1299 cell lines.

(A) Western blot analysis of epithelial markers E-cadherin, mesenchymal markers (N-cadherin, Vimentin) and Snail in A549 cells with different Keap1 expression levels. (B) Quantitative analysis of EMT protein in A549 cell line was shown. (C) Western blot analysis of epithelial markers E-cadherin, mesenchymal markers (N-cadherin, Vimentin) and Snail in H1299 cells with different Keap1 expression levels. (D) Quantitative analysis of EMT protein in H1299 cell line was shown. Statistical significance was tested by unpaired t test. Values were given as mean ± SD. *P < 0.05, **P < 0.01, ***P < 0.001 compared to the control group.

Keap1 knockdown would lead to the chemotherapy resistance in A549 and H1299 cell lines

To explore the Keap1 role on chemoresistance, we treated A549 and H1299 cells lines with different concentrations of cisplatin and paclitaxel respectively. The cell viability of cell lines decreased with increasing drug concentration (Figs. 6A and 6C). Knockdown of Keap1 promoted cell viability and resisted A549 and H1299 cells to cisplatin and paclitaxel, while overexpression of Keap1 showed the opposite result (Figs. 6A and 6C). Cell viability was assessed using the CCK8 assay. In addition, the IC50 of cisplatin and paclitaxel were reduced in both the cell lines over expressed Keap1 compared to the control cells, which could be reversed by Keap1 knockdown (Figs. 6B and 6D). These results indicated that Keap1 may confer chemotherapy sensitivity in lung cancer cells.

Figure 6 Keap1 knockdown confered the chemotherapy resistance in A549 and H1299 cells.

(A) A549 and H1299 cell liens expressing different levels of Keap1 were treated with different concentrations of cisplatine, and the cell viability was determined by CCK8 assay. (B) IC50 of A549 and H1299 cell lines expressing different levels of Keap1 to cisplatin. (C) A549 and H1299 cell liens expressing different levels of Keap1 were treated with different concentrations of paclitaxel, and the cell viability was determined by CCK8 assay. (D) IC50 of A549 and H1299 cell lines expressing different levels of Keap1 to paclitaxel. Statistical significance was tested by unpaired t test. Values were given as mean ± SD. *P < 0.05, **P < 0.01 compared to the control group.

Keap1 knockdown promoted Nrf2 nucleus translocation

Keap1 is a major negative regulator of Nrf2, which has been shown to inhibit the proliferation and cisplatin resistance of tumor cells. We therefore hypothesized that when the expression level of Keap1 was abnormal, the expression level of Nrf2 would change accordingly. We then compared the protein expression levels of total Nrf2 (T-Nrf2), nucleus Nrf2 (N-nrf2), and Heme Oxygenase-1 (HO-1) regulated by Nrf2 in Keap1 KD or OE cell lines, respectively, and found that both T-Nrf2 and N-Nrf2 were reduced upon Keap1 overexpression, and downstream HO-1 was also reduced, while knockdown of Keap1 showing the opposite results (Figs. 7A and 7B). The results were further confirmed by treating tumor cells with a certain concentration of H2O2 (A549 200 µM, H1299 100 µM) for 30 min (Figs. 7C and 7D).

Figure 7 Keap1 knockdown promoted Nrf2 into the nucleus of tumor cells.

(A) Western blot analysis of total Nrf2 (T-Nrf2), HO-1 and nucleus Nrf2 (N-Nrf2) in two cells with different Keap1 expression levels. (B) Quantitative analysis of T-Nrf2, HO-1 and N-Nrf2 proteins in A549 and H1299 cell lines. (C) Western blot analysis of T-Nrf2, HO-1 and N-Nrf2 in cells with different Keap1 expression levels under oxidative stress (H2O2) conditions. (D) Quantitative analysis of T-Nrf2, HO-1 and N-Nrf2 proteins in cells with different Keap1 expression levels under oxidative stress (H2O2). Statistical significance was tested by unpaired t test. Values were given as mean ± SD *P < 0.05, **P < 0.01, ***P < 0.001 compared to the control group.

Discussion

The antioxidant stress system protects normal cells from ROS and also creates favorable conditions for the survival of tumor cells. The tumor microenvironment (TME) is a region where tumor inflammation, hypoxia and oxidative stress coexist. and is a well-recognized actor in the development of cancer and adaptation of tumor cells to therapy (Hinshaw & Shevde, 2019; Paardekooper, Vos & van den Bogaart, 2019). The Keap1-Nrf2 signaling pathway is the most important anti-oxidative stress pathway in vivo, and its aberrant activation has been found in various tumors, such as esophageal cancer and colon cancer (Zhang et al., 2018; Sadeghi et al., 2017). At present, it is generally believed that aberrant activation of Nrf2 promotes radioresistance and chemoresistance of various cancers (Leinonen et al., 2015). However, as the master inhibitor of Nrf2, the role of Keap1 in tumors is controversial. On the one hand, Keap1 plays a role similar to that of oncogenes, and on the other hand, Keap1 abnormalities promote the biological malignant effects of tumor cells and are associated with poor prognosis in solid tumors (Fabrizio et al., 2017; Fabrizio et al., 2019).

In the present study, we utilized IHC analysis to detect the expression of Keap1 in 37 surgically resected LUADadenocarcinoma and adjacent tissues. We identified that Keap1 expression in cancerous tissues was statistically lower compared to adjacent tissues. Keap1 expression was inversely correlated with the clinical stage and lymph node metastasis and early-stage tumors (I) without lymph node metastasis had higher levels of Keap1 expression compared with late-stage tumors (II and III). But the expression of Keap1 was independent of age, gender, smoking history, tumor location, degree of differentiation and tumor maximum diameter. Our results are consistent with previous research (Chien et al., 2015). These findings suggest that Keap1 may be involved in tumor progression, which could provide a new target for intervention in the treatment of LUAD.

Keap1 is important in protection of oxidative stress. Recent studies have shown that Keap1 affects tumor malignant behaviors. Overexpression of Keap1 in glioma cells decreased their migratory and proliferative behaviors while knockdown of Keap1 increased this abrogating behavior (Fan et al., 2017). These phenomena are consistent with those observed in breast cancer cells (Zhang et al., 2019). Animal experiments further indicated that Keap1 R320Q somatic mutation could accelerate tumor growth in nude mouse (Gong et al., 2020). In this study, we observed that high levels of Keap1 expression significantly correlated with lower proliferation and migration of LUAD cells and Keap1 knockdown facilitated EMT of tumor cells. We hypothesized that this may be related to the effect of Nrf2 expression. It has been reported that Nrf2 nuclear translocation could activate Notch1/Snail signaling pathway, which accelerated EMT and metastasis of tumor cells (Han et al., 2020). In addition, PDCD4 inhibits the EMT pathway in tumors by increasing Keap1 expression and decreasing the level and activity of Nrf2 (Hwang, Jeong & Chang, 2020), our study also found that Keap1 knockdown promoted Nrf2 nuclear translocation. However, the specific pathways by which Keap1 affects metastasis of LUAD cells require further mechanistic studies. These observations suggest that Keap1 protects lung cancer proliferation and metastasis and may serve as a biomarker for LUAD.

Chemotherapy resistance effectively blocks lung cancer treatment efficacy. It has been reported that Nrf2 contributed to chemotherapy resistance of tumors. LncRNA TUG1 increases cisplatin resistance in prostate cancer and esophageal cancer by the Nrf2 signal axis (Yang et al., 2020; Zhang et al., 2019). Nrf2 knockdown enhances sensitivity to cytarabine and daunorubicin in acute myelogenous leukemia cell lines and inhibits tumor growth in a mouse model of myelodysplastic syndrome (Karathedath et al., 2017; Lin et al., 2019). As a master inhibitor of Nrf2, Keap1 is inextricably linked to chemoresistance. In the present study, we treated aberrant Keap1 expression in A549 and H1299 cell lines with different concentrations of chemotherapy drugs and founded that overexpression of Keap1 may conferred the chemotherapy sensitivity in tumors.

In conclusion, this study investigated the effect of Keap1 on LUAD growth and drug resistance more systematically through clinical and cellular experiments. The results showed that Keap1 was lowly expressed in LUAD. The high expression level of Keap1 protein inhibited the growth and metastasis of LUAD cells and enhanced the sensitivity to chemotherapy, suggesting an anti-tumor effect of Keap1. It is speculated that the possible reason for this effect is related to the action of Keap1 on Nrf2. Overexpression of Keap1 promotes Nrf2 degradation and reduces Nrf2 nuclear translocation, which affects the subsequent anti-oxidative stress effect and disrupts the balance of TME. Ultimately, it inhibits tumor cell growth and metastasis, decreases the expression of antitumor drug efflux-related proteins and glutathione levels, and enhances cellular sensitivity to chemotherapy sensitivity.

Conclusions

Our conclusion confirmed the involvement of Keap1 in LUAD progression and drug sensitivity. Keap1 is expected to serve as a potential tumor marker to guide the staging and treatment of lung cancer. However, there are still limitations: first, the relatively small sample size may lead to selection bias, and the lack of clinical response of patients to chemotherapy may weaken the validity of the results. Second, the effect of Keap1 on LUAD is limited to the cellular phenotype, and further investigation of its molecular mechanisms is essential.

Supplemental Information

Supplemental Information 1 Clinical data.

Click here for additional data file.

Supplemental Information 2 Figure 1 raw data.

The immunohistochemistry assay and Keap1 mRNA in different lung cancer cell lines and HBE by RT-qPCR in Fig. 1.

Click here for additional data file.

Supplemental Information 3 Figure 2 raw data.

Keap1 mRNA in A549 and H1299 cell lines in Fig. 2.

Click here for additional data file.

Supplemental Information 4 Figure 3 raw data.

Cell growth curves and colony formation assays in Fig. 3.

Click here for additional data file.

Supplemental Information 5 Figure 4-1 raw data.

Transwell invasion assays in Fig. 4.

Click here for additional data file.

Supplemental Information 6 Figure 4-2 raw data.

Wound scratch tests in Fig. 4.

Click here for additional data file.

Supplemental Information 7 Figure 6 raw data.

Cisplatin and paclitaxel treatment and drug sensitivity assay in Fig. 6.

Click here for additional data file.

Supplemental Information 8 Uncropped blots.

Western blot analysis in Figs. 1 and 2.

Click here for additional data file.

Supplemental Information 9 Codebook.

Western blot analysis in Figs. 5 and 7.

Click here for additional data file.

We wish to thank Mr. Lun Zhu for his assistance in collecting clinical specimens.

Additional Information and Declarations

Competing Interests

Author Contributions

Human Ethics

Data Availability

The authors declare that they have no competing interests.

Hong Gao conceived and designed the experiments, performed the experiments, analyzed the data, prepared figures and/or tables, and approved the final draft.

Peipei Tang conceived and designed the experiments, performed the experiments, analyzed the data, prepared figures and/or tables, and approved the final draft.

Kejie Ni analyzed the data, prepared figures and/or tables, and approved the final draft.

Lun Zhu analyzed the data, prepared figures and/or tables, and approved the final draft.

Song Chen analyzed the data, authored or reviewed drafts of the paper, and approved the final draft.

Yulong Zheng conceived and designed the experiments, authored or reviewed drafts of the paper, and approved the final draft.

Yufeng Wan conceived and designed the experiments, authored or reviewed drafts of the paper, and approved the final draft.

The following information was supplied relating to ethical approvals (i.e., approving body and any reference numbers):

The study was approved by the Institutional Review Board of the Affiliated Huai’an Hospital of Xuzhou Medical University (HEYLL201920).

The following information was supplied regarding data availability:

The raw clinical data are available in the Supplemental File.

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
