# Peer review of "Inhibition of Kelch-like epichlorohydrin-related protein 1 promotes the progression and drug resistance of lung adenocarcinoma"

_PeerJ, doi:10.7717/peerj.11908_

## Round 0.1 · original submission · Major Revisions

First of all, I appreciate the authors' endeavor on in vitro and clinical studies to examine the association between the malignancy and drug resistance in lung cancer and the expression level of Keap1. Please read the reviewers' comments and address the answers.

In addition to the reviewers' comments, I would like to remark several points on the manuscript as follows.

- There are preexisting data banks that are publicly available including TCGA as the reviewer mentioned or COSMIC. It would be meaningful to utilize these databases to compare the current findings on Keap1.

- I assume that there was reasoning to select two cell lines out of four, knowing that A549 possesses mutant Keap1 (ref.34) with a lower expression level whereas H1299 expresses wild-type Keap1 (ref.34) with a higher expression level among lung cancer cells examined (Fig.1). Please elaborate on the rationale for choosing these cell lines.

-In regard to the mutant type of Keap1 in A549, could you confirm if it is related to its function (e.g. reduced binding affinity to Nrf2 leading to the increased translocation of Nrf2, etc.)? Also, please clarify whether Nrf2 in both cell lines (A549 and H1299) serves normally as a transcription factor.

- It seems that H1299 is more malignant than A549 according to Fig1C (higher proliferation) and 4D (greater invasiveness). Also, the impact of Keap1 knockdown looks greater in H1299 than in A549. Would it be possible to explain the reason behind it?

- In Fig.5, an IC50 value in A549 (Ctl) is higher than that in H1299 (Ctl). Could you add the reference if there is any study on the reduced potency of cisplatin in A549 cell viability than H1299? Could you double-check that there are research articles on a correlation between mutant Keap1 and sensitivity toward cisplatin?

-Mistypings are found frequently. For instance, alphabet u instead of micro in Greek, box instead of less than sign in p-value, extra spaces, etc. Please correct those accordingly.

Reviewer 1 ·

Basic reporting

In the manuscript entitled “Inhibition of Kelch-like epichlorohydrin-related protein 1 promotes the proliferation and drug resistance of lung adenocarcinoma”, the authors illustrated that Keap1 knockdown promotes tumor cell growth, proliferation, invasion, metastasis and chemotherapy resistance in lung adenocarcinoma. There are a few comments that should be addressed to further improve the manuscript.

Experimental design

1. The authors should consider including the public database like the TCGA database to demonstrate the Keap1 level difference between patients and healthy control in figure 1.

2. In figure 3, the authors demonstrated that Keap1 knockdown increased cell proliferation. However, in figure 4, Keap1 knockdown increased cell migration and invasion. Please clarify if the cell migration and invasion is in fact directly affected by Keap1 knockdown, rather than caused by the increased cell proliferation.

Validity of the findings

3. What is the rationale to choose A549 and H1299 cell lines for in vitro study? The authors should discuss this in the manuscript. Moreover, in figure 1, these two cell lines showed different Keap1 level, which seems as if these two cell lines have different cell capabilities. However, both cell lines have shown similar results in figure 2,3,4, and 5, which should be further clarified.
4. There are no scale bars in figure 4A and 4C. Please update the figures with scale bars for clarity.

Reviewer 2 ·

Basic reporting

The authors provide sufficient background and the article is well structured with detailed supporting data leading to addressing the questions asked in the hypothesis.

Experimental design

Experiments are detailed and address the aims of the proposed study. However, no data has been presented to understand some of the key aspects of Keap1 regulation. Some of the major questions/concerns I had while going through the manuscript:

1. Loss of Keap1 expressions in H460 and H1299 cell lines (Fig 1E) are not robust. SE bar not shown in the HBE control data.

2. The authors should verify if similar observations as shown in Tables 1 and 2 are found, and an inverse relationship exists between Keap1 expressions and LUAD in patient data from the TCGA database.

3. What were the mutational states in the keap1 gene in the clinical LUAD tissues? Were any common mutational patterns observed in the high and low expression groups?

4. What were the expression levels of Nrf2 in the (i) LUAD tissues compared to the non-cancerous tissues and (ii) in the tested cell lines alone compared to those with Keap1 KD or OE? Can be assessed by Westerns/ IHC and mRNA levels. Correlations with Keap1?

5. Did aberrant Keap1 levels affect Nrf2 nuclear translocation in the LUAD cells?

6. Minor typing errors, for example line 189- replace “hanman” with “human”; line 317- “limitations” in place of “laminations”.

Validity of the findings

The experimental approach for the given hypothesis is straight forward, however the findings in this study lack novelty. The anti-tumorigenic role of Keap1 in lung cancer is well studied and established. Studies have also shown that loss of keap1 results in cisplatin resistance in lung cancer.

Discussion is more speculative and does not really connect the findings shown in the study. Reorganization of the discussion with major focus on the findings from this study is warranted.

Reviewer 3 ·

Basic reporting

Gao and colleagues investigate the effect of Keap1 inhibition/overexpression on cell proliferation, migration/invasion and drug resistance providing some interesting findings in two lung adenocarcinoma cell lines (A549 and H1299). To support these claims authors use effective knockdowns and overexpression constructs and provide a series of functional experiments.

If the authors can address the following minor concerns I would support publication of this interesting concept.

Minor revisions:


1) In Figure 2, using the lysates of the Keap1 KD/OE cells could the authors test the expression levels of Nrf2 or some of its target genes? This should be also performed after treatment with H2O2. (oxidative stress conditions)
2) In Figure 4, the authors claim that Keap1 inhibition promotes invasion and migration but they do not show differences in the expression of EMT markers.
3) In Figure 5, can the authors extend their observations on cisplatin resistance using a second chemotherapeutic drug, for example paclitaxel?


Minor concerns/Suggestions

The authors should provide a concise and accurate background of the keap1-Nrf2 complex in the introduction section instead of the discussion, where they should summarize their main findings, emphasize their novelty, compare them with similar results from other experimental groups and discuss possible translational applications.
Example: lines 236-277 should be removed to the introduction section.


Lines 203-210: The authors should provide the specific number of the figure legend when they describe their results.

Line 311: the phrase “keap1 overexpression has a beneficial effect on chemoresistance” is confusing.

Experimental design

Some of the findings as mentioned in the basic reporting should be further confirmed with additional experiments.

Validity of the findings

No comment.

---

## Round 0.2 · accepted · Accept

In my view, all critiques were adequately addressed, and the revised manuscript is acceptable.

Reviewer 2 ·

Basic reporting

Authors have addressed majority of my comments. I would recommend incorporating the observations and figures generated from TCGA datasets either in the main figures or supplement with supporting text in the results section.

Experimental design

Sound.

Validity of the findings

No comments.

Reviewer 3 ·

Basic reporting

The authors have addressed all my concerns and therefore I support the publication of this manuscript.

Experimental design

No comment

Validity of the findings

No comment

Additional comments

There are still some typo and grammatical errors in the manuscript that need to be corrected.